



# DL-RMD: A geophysically constrained electromagnetic resistivity model database for deep learning applications

Muhammad Rizwan Asif[1,3,4], Nikolaj Foged[1,4], Thue Bording[1,2], Jakob Juul Larsen[3,4], and Anders Vest Christiansen[1,4]

[1]Hydro-Geophysics Group (HGG), Department of Geoscience, Aarhus University, Aarhus C, 8000, Denmark.
[2]Aarhus GeoInstruments, Åbyhøj, 8230, Denmark.
[3]Department of Electrical and Computer Engineering, Aarhus University, Aarhus N, 8200, Denmark.
[4]Aarhus University Centre for Water Technology (WATEC), Aarhus N, 8200, Denmark.

*Correspondence to*: Muhammad Rizwan Asif (rizwanasif@geo.au.dk, rizwansheikh123@hotmail.com)

**Abstract.** Deep learning algorithms have shown incredible potential in many applications. The success of these data-hungry methods is largely associated with the availability of large-scale data sets, as millions of observations are often required to achieve acceptable performance levels. Recently, there has been an increased interest in applying deep learning methods to geophysical applications where electromagnetic methods are used to map the subsurface geology by observing variations in the electrical resistivity of the subsurface materials. To date, there are no standardized datasets for electromagnetic methods, which hinders the progress, evaluation, benchmarking, and evolution of deep learning algorithms due to data inconsistency. Therefore, we present a large-scale electrical resistivity model database of a wide variety of geologically plausible and geophysically resolvable subsurface structures for the commonly deployed ground-based and airborne electromagnetic systems. The presented database can potentially be used to build surrogate models of well-known processes and aid in labour intensive tasks. The geophysically constrained property of this database will not only achieve enhanced performance and improved generalization but, more importantly, it will incorporate consistency and credibility in deep learning models. We show the effectiveness of the presented database by surrogating the forward modelling process, and urge the geophysical community interested in deep learning for electromagnetic methods to utilize the presented database. The dataset is publically available at https://doi.org/10.5281/zenodo.7260886 (Asif et al., 2022a).

## 1 Introduction

Recent years have witnessed the success of many deep learning (DL) applications. Although, DL emerged in 1982 in the form of neural networks (Hopfield, 1982), it started to gain attention in 2012 due to its notable performance for image classification tasks (Krizhevsky et al., 2017, 2012). Since then, it has been successfully applied for many applications including object





detection (Asif et al., 2019; Redmon et al., 2016; Ren et al., 2015), image super-resolution (Dong et al., 2016; Zhang et al.,

2018), speech recognition (Zhang et al., 2017), and stock market predictions (Pang et al., 2020). The revival of DL was mainly influenced by the availability of cheap computing resources, deeper network architectures and large-scale publically available datasets. Deeper network architectures and increased number of samples in the training datasets are key factors for improved performance and better generalization of DL models (Wang et al., 2016).

Geophysics is a branch of Earth sciences, and geophysical methods are often used to infer information about the subsurface

geology by mapping physical properties. The integration of neural networks in geophysics started several decades ago and has covered many domains of geophysics (Baan and Jutten, 2000; Dramsch, 2020) including seismic (Röth and Tarantola, 1994; Zhang et al., 2020), magneto-telluric (Conway et al., 2019; Liu et al., 2020; Zhang and Paulson, 1997) geo-mechanical (Feng and Seto, 1998; Khatibi and Aghajanpour, 2020) and electromagnetics (Birken and Poulton, 1999; Birken et al., 1999; Bording et al., 2021; Kwan et al., 2015; Poulton et al., 1992; Zhu et al., 2012). Interestingly, the last few years have seen a significant

increase of interest in applying DL to electromagnetic (EM) methods (see Table 1), where the artificially generated EM fields are used to map variations in the electrical resistivity properties of the subsurface. For more details regarding the EM methods, the readers are referred to literature, e.g. (Kirsch, 2006). The increasing interest in applying DL to EM methods is mainly influenced by the increased ability of the EM methods to collect huge data sets in short amounts of time, which makes the subsequent processes extremely laborious and time consuming. Therefore, a DL method could be beneficial in surrogating

well-known EM processes, e.g. forward modelling where the propagation of the EM fields are simulated resulting in the forward responses (Xue et al., 2020), and inverse modelling (inversion) where the electrical resistivity properties of the subsurface are deduced from observed EM data (Zhdanov, 2015). DL methods can also assist in manual tasks, which may require considerable time when performed manually, such as anomaly detection in EM data. Further opportunities may lie in other tasks, e.g. data de-noising.

To apply a DL algorithm to EM methods for various applications, subsurface resistivity models and/or the corresponding EM responses are often required. To achieve optimal performance, a DL method should be trained on a large number of geologically realistic subsurface models. Evident from Table 1, the recently developed DL methods either uses subsurface resistivity models acquired from field data or generate the models randomly or in a pseudorandom manner for training. However, a method trained on random models, where the resistivity of each geological layer is chosen from a probability

distribution, would not result in optimal performance as many of the training samples would be geologically unrealistic. A good solution is either using resistivity models inverted from field data or pseudorandom resistivity models where the resistivity of the training models are based on some prior geological information to reflect various characteristics of field data (Bai et al., 2020). However, a DL method trained on such training samples would only be effective for specific geological conditions and may result in unsatisfactory performance for significantly different geological settings, as bias in the training



data can substantially affect generalizability. Additionally, the unavailability of standard benchmark database hinders the progress, evaluation, benchmarking, and evolution of DL algorithms due to data inconsistency (Bergen et al., 2019; Reichstein et al., 2019).

| Reference | No. of samples in training set | Training Observation type | Application |
|---|---|---|---|
| (Wu et al., 2021a) | 80,000 | Pseudorandom resistivity models and forward responses | Inversion |
| (Colombo et al., 2021a) | 5,000 | Pseudorandom resistivity models and forward responses | Inversion |
| (Colombo et al., 2021b) | 20,000 | Random resistivity models and forward responses | Inversion |
| (Wu et al., 2021b) | 16,800 | Forward responses of random resistivity models | De-noising |
| (Bording et al., 2021) | 93,500 | Field data and inversion models | Forward modelling |
| (Puzyrev and Swidinsky, 2021) | 5,12,000 | Random resistivity models and forward responses | Inversion |
| (Asif et al., 2021a) | 100,000 | Field data and inversion models | Forward modelling |
| (Moghadas et al., 2020) | 20,000 | Random resistivity models and forward responses | Forward modelling |
| (Bai et al., 2020) | 12,000 | Pseudorandom resistivity models and forward responses | Inversion |
| (Li et al., 2020) | 1,000,000 | Pseudorandom resistivity models and forward responses | Inversion |
| (Bang et al., 2020) | 25,173 | Pseudorandom resistivity models and forward responses | Inversion |
| (Noh et al., 2020) | 20,000 | Random resistivity models and forward responses | Inversion |
| (Moghadas, 2020) | 20,000 | Random resistivity models and forward responses | Inversion |
| (Colombo et al., 2020a) | 2,35,620 | Pseudorandom resistivity models and forward responses | Inversion |
| (Colombo et al., 2020b) | 88 | Pseudorandom resistivity models and forward responses | Inversion |
| (Lin et al., 2019) | 2,400 | Field data and inverted model forward responses | De-noising |
| (Guo et al., 2019) | 10,000 | Pseudorandom resistivity models and forward responses | Inversion |
| (Puzyrev, 2019) | 20,000 | Pseudorandom resistivity models and forward responses | Inversion |
| (Qin et al., 2019) | 50,000 | Random resistivity models and forward responses | Inversion |

**Table 1: Recent publications (2019-2021) of DL in EM which shows the number of models and/or forward responses in the training dataset and the type of training dataset which are either random, pseudorandom or models inverted from field data**

To have an inclusive DL solution for various applications in EM, we present a physics-driven large-scale model database (~1 million) of geologically plausible and EM resolvable 1-D sub-surface resistivity models spanning the resistivity range from 1 Ωm to 2000 Ωm and to a depth of 500 m. This model database is suitable for ground-based and airborne EM systems in a DL context. We use broad-banded von Kármán covariance functions to generate geologically constrained resistivity models. Geophysical constraints are imposed by calculating the EM forward data of the initial resistivity models followed by inversion 70 of the EM forward to obtain the final resistivity models. This allows us to create a comprehensive resistivity model database (RMD) that may not only improve performance and generalization, but would also incorporate consistency and reliability in the DL models. We believe that the presented RMD will be a valuable resource to accelerate the inter- and trans-disciplinary research of Earth and data sciences. The presented DL-RMD will also provide uniformity in training and benchmarking for DL methods in EM. Therefore, we urge the geophysical community interested in DL for EM methods to use the DL-RMD.





This rest of the paper is organized as follows: Section 2 describe the general methodology of generating the sub-surface resistivity models, while specific settings for the DL-RMD for the three EM system categories is specified in section 3. Section 4 provide details for training a DL method to surrogate the forward modelling problem and show the effectiveness of the DL-RMD. Discussion, code and data availability, and concluding remarks are given in section 5, 6, and 7 respectively.

## 2 Methodology

Geological processes do not result in random structures, nor are the subsurface resistivity structures random, as some spatial correlation is generally present (Tacher et al., 2006). Therefore, it is reasonable that the training of a DL method is based on subsurface structures that are geologically plausible and in an EM context, over all resolvable by the EM method. Additionally, the scale of the resistivity structure in the models should reflect the resolution capability of the EM methods, as training a DL method to resolve structures that are not evident in the input data is not possible. EM method are diffusive methods with
significantly decreasing resolution with depth and the electrical conductivity contrast plays an important role for the resolution capability; hence, a metric number for a given EM method's resolution capability and the depth of investigation can not be given.

To obtain geologically realistic models, we use the broad-banded von Kármán covariance functions (Møller et al., 2001) to generate geologically plausible models (von Kármán models). The suite of von Kármán models consist of fine geological
structures, and contain some resistivity variations and patterns that are unlikely to be resolved due to the resolution limitation of the EM method. To replicate the resolution capability of the EM method, we generate EM forward responses of the initially over-detailed von Kármán models and invert these forward responses to obtain the final resistivity models. Since we aim at generating 1-D resistivity models, we are only concerned about the resistivity ($\rho$) variations in the vertical direction (z) from surface to some depth in our model generation.

Initially, we base the spatial variation character of (z, log($\rho$)) for our von Kármán models on the broad-banded von Kármán covariance functions (Christiansen and Auken, 2003; Møller et al., 2001).

$$C(z, A, v) = A^2 C_0 \left(\frac{z}{L}\right)^v K_v \left(\frac{z}{L}\right),$$
(1)

where $A$ becomes the amplitude of the logarithmic resistivity, $C_0$ is a scaling constant, $Z$ the spatial (vertical) distance, $L$ characterizes the maximum correlation length accounted for, and $K_v$ is the modified Bessel function of second kind and order
v. In the model generation, $L$ is fixed to a high number (1800 m) which gives us strong correlation for z<<$L$ (Maurer et al., 1998). By using combinations of v, $C_0$, and resistivity, and compiling several realizations of the stochastic von Kármán process, we generate a variety of resistivity models on multiple scales. Table 2 summarizes the $L$, v, $C_0$, and resistivity values used.



| Parameter | Values |
|---|---|
| Resistivity | 1 to 2000 Ωm, log spaced, 20 values per decade |
| $L$ | Fixed: 1800 m |
| $v$ | [0.6, 0.7, 0.8, 0.9, 1.0] |
| $C_0$ | [0.5, 1, 2, 4] |
| # sharp boundaries | [1, 2, 3, 4, 5] |

**Table 2: Parameters used in all combinations to generate the initial von Kármán resistivity models.**

Examples of this are shown in Figure 1(a-c) where the von Kármán models (in black curves) are generated with a combination
of the extreme values of $v$, $C_0$ for an initial resistivity value of 30 Ωm. Low $v$ and high $C_0$ produce models with fine and large-
scale variations (Figure 1a), while high $v$ and high $C_0$ values produce a relatively smooth model (Figure 1b) but still with
resistivity variations spanning 2-3 decades of resistivity. The combination of low $v$ and $C_0$ values ensure that the simple and
close to half-space models are also represented (Figure 1c).

Sharp layering in subsurface are plausible, and large resistivity amplitudes and short correlation lengths in the von Kármán
functions will form layering in the models. To include more models with sharp layering, we stitch 2-6 randomly selected depth
intervals of the initial generated von Kármán models from a uniform distribution. An example of a stitched model is shown in
Figure 1(d). These stitched models also ensure that different combination of $v$, $C_0$, are represented within one model.

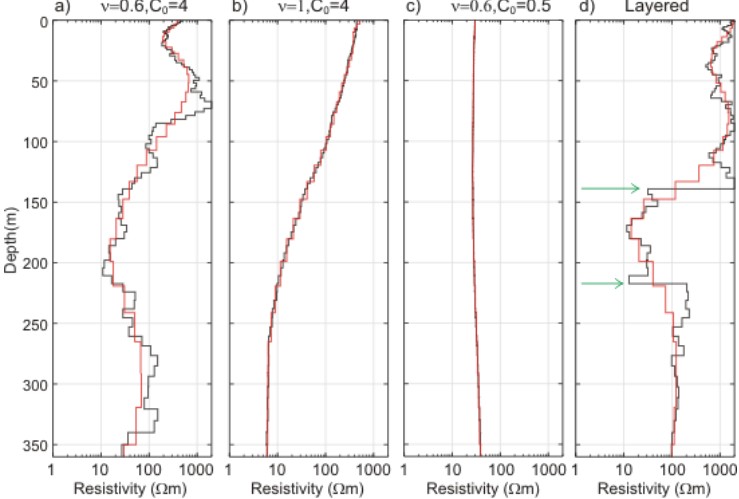

**Figure 1: Examples of von Kármán models and the result after the forward and inversion process, where black curves shows von
Kármán models (re-discretized to 90 layer) and red curve shows the final model. (a-c) for the combination of $v$, $C_0$ stated in the title,
(d) a stitched layered model (green arrows marks the imposed sharp layer boundaries). The red curves show the obtained model
from inversion of the forward response of the black model.**



Prior to the EM forward calculation, the von Kármán models are re-discretized to 90-layers for faster forward computation and easier handling. For the forward calculation, the geometric mean of the last 5 meters of the von Kármán models is assigned to the last model layer that continues to infinity depth. The calculated forward data are assigned a uniform uncertainty of 5% and is inverted with a 30-layer model with a minimum structure (smooth) regularization scheme (Viezzoli et al., 2008). The red model curves in Figure 1 are the resistivity models after the forward and inversion process, and are the models that enter

the DL-RMD. As seen from Figure 1, the von Kármán models hold structures that are not resolved by the inverted resistivity models, so the models obtained after the forward and inversion process results in structures resolvable by the EM method. A total of ~95% of the inverted resistivity models explain (fits) the forward data within the assumed data uncertainty.

The forward and inverse modelling is carried out for three different generic time-domain EM (TEM) systems spanning different depth range using the AarhusInv modelling code (Auken et al., 2015). The specific RMD settings for different TEM systems

are summarized in section 3.

## 3 Deep learning resistivity model database (DL-RMD)

EM systems for subsurface exploration have existed since the 1950s, and nowadays a large variety of airborne and ground-based time-domain electromagnetic (TEM) and frequency-domain electromagnetic (FEM) systems exist. Both TEM and FEM methods map the electrical resistivity of the subsurface by inducing EM fields. TEM methods records the decay of the

secondary EM-field, in the absence of the transmitted EM-field in time-domain, while FEM methods records the secondary EM-field in frequency-domain in the presence of the transmitted EM-field (Christiansen et al., 2006). TEM and FEM methods also differ in resolution and depth of investigation (DOI), depending on the TEM-system configuration, e.g. transmitter turn-off time, transmitter moment, airborne/ground-based. For the RMD to be compatible for different TEM systems, we have compiled three model databases with ~1 million models in each for three generic TEM-systems with different DOI as their

primary differences. We refer to the three RMD as *shallow, intermediate*, and *deep*, with the acronyms S-RMD, I-RMD, D-RMD respectively.  S-RMD mimics a shallow focusing ground based TEM system, initiated by a short transmitter turn-off time. For S-RMD, the models are discretized down to 125 m with a top layer thickness of 0.5 m. I-RMD and D-RMD mimics airborne TEM with different DOIs, hence discretized down to a depth of 350 and 500 m and with a top layer thickness of 3 and 5 m respectively.The calculation of DOI follows (Christiansen and Auken, 2012).

The model discretization for three RMD for the initial von Kármán models and for the final resistivity models entering the RMD are summarized in Table 3. Table 3 also hold the key specifications of the three generic TEM systems. The settings for the generation of the von Kármán models have been specified in Table 2 and are common for the three RMD. Each of the three RMD holds ~1 million models spanning the resistivity interval 1-2000 Ωm, where 1/6 of the models originate from the initial generated von Kármán models and 5/6 of the models come from the stitched layered von Kármán models.



| Type | Parameter | S-RMD | I-RMD | S-RMD |
|------|-----------|-------|-------|-------|
| **von Kármán models** | Max depth (m) | 125 | 355 | 505 m |
| | Discretization (m) | 0.1 | 0.1 | 0.1 |
| | Re-discretization (m) | 0.2-120 m, 90 layer log-spaced | 1-350 m, 90 layer log-spaced | 2-500 m, 90 layer log-spaced |
| **Database resistivity Models** | Model Discretization | 0.5-120 m, 30 layer log-spaced | 3-350 m, 30 layer log-spaced | 5-500 m, 30 layer log-spaced |
| **Generic TEM configuration** | Turn off time (µs) | 4 | 12 | 40 |
| | *Gate time start (µs) | 5 | 13 | 50 |
| | *Gate time end (ms) | 1 | 10 | 32 |
| | Modelling high (m) | 0 – Ground-based | 40 - Airborne | 40 - Airborne |

**Table 3: Model discretization and key specifications of the generic TEM systems for three resistivity model databases. The generic TEM system are all central loop configurations. *Gate start/end times has zero-time reference at begin of turn-off time.**

Some insights on the three RMD are given in Figure 2, where Figure 2(a-c) shows the layer resistivity distribution of the three RMD. The resistivity distribution of the von Kármán models were generated uniformly, but the forward and inversion process makes the resistivity distribution slightly skewed towards the lower resistivity end, due to the lower sensitivity/resolution in

the high resistivity end for the EM method (Christiansen et al., 2006; Jørgensen et al., 2005). The larger start and end bins compared to the neighboring bins in Figure 2(a-c) are due to the 1 Ωm and 2000 Ωm resistivity truncation. The estimated DOI for the three RMD are shown in Figure 2(d-f). We observe that approximately 70% of the models have DOI less than the bottom to last layer boundary of the given RMD. Especially, a thick conductive layer near the surface will significantly limit the DOI for a given TEM configuration. The uneven and, in some cases, limited DOI does not pose a problem for a DL

algorithm, as the EM method will compromise a similar DOI limitation for the given resistivity model (see discussion section for more details).



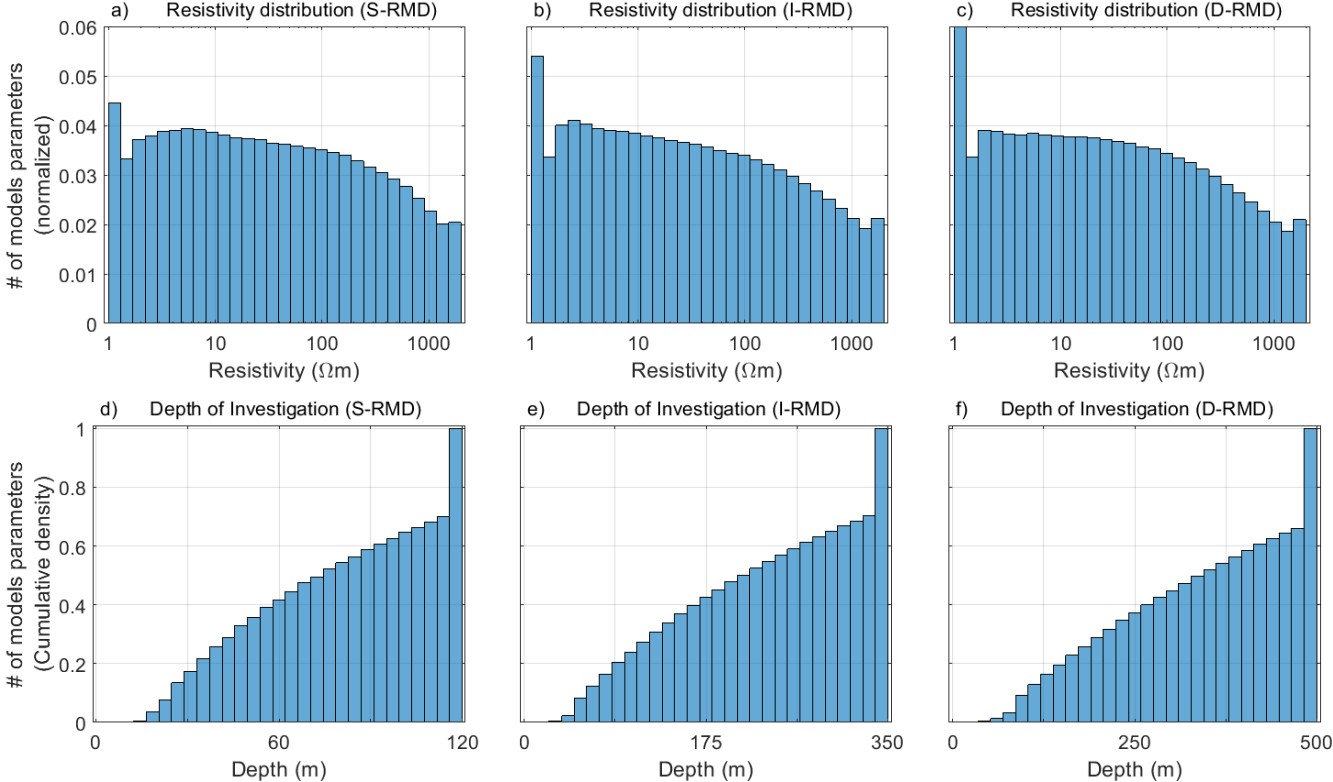

**Figure 2: Statistical insights on the DL-RMD. (a-c) Resistivity distribution of the S-RMD, I-RMD, and D-RMD respectively. (d-f) Distribution of depth of investigation of models in the S-RMD, I-RMD, and D-RMD respectively, plotted as cumulative sum.**

## 4 Example of an EM application using RMD

EM methods can benefit from the presented RMD in many ways. For example, the RMD can be used to surrogate the computationally expensive numerical forward modelling by using a computationally efficient DL method which would speed up the whole inversion process. It can also be used to develop a DL algorithm to replace the calculation of the partial derivatives in deterministic inversion methods where the subsurface resistivity model is updated iteratively by using the partial derivatives of the model parameters. Detecting anomalies in the EM data by using a DL approach using the RMD can significantly speed-up the EM data processing and limit the involvement of human-centric manual workflows. Additionally, EM data de-noising also becomes plausible.

As an example in this paper, we use the RMD to surrogate the forward modelling problem for a ground-based TEM system using a fast DL method since a significant number of forward calculations are required during the inversion process, when either deterministic or stochastic inversion methods are used. By replacing the computationally expensive numerical forward



modelling approach, the whole inversion process may be accelerated without further modification to a standard inversion workflow (Asif et al., 2021b). However, it is crucial that the performance of the network balances the numerical precision and increased speed of computation. If the prediction accuracy is not sufficiently high, the application in an inversion framework

may result in spurious subsurface features and erroneous geological interpretations of the geophysical EM mapping results.

## 4.1 Deep learning (DL) setup

We design the surrogate model for the tTEM system (Auken et al., 2018). The tTEM system is a ground-based towed TEM system with a maximum depth of investigation of 120 m based on the data time interval from ~5 µs to ~1 ms, which matches the specification of S-RMD, therefore, we use it to train our DL method.

The input to the DL algorithm becomes the 30-layer resistivity model $\mathbf{m}$ in S-RMD, where the layer thickness of each resistivity layer is fixed. The target outputs are the numerical TEM forward responses, i.e. $\frac{d\mathbf{B}}{dt}$, for the corresponding inputs. A standard EM modelling code (Auken et al., 2015) is used to generate the TEM forward responses for the resistivity models $\mathbf{m}$ with fixed layer thicknesses. We generate the responses from ~1 ns to ~10 ms with exponentially increasing gate-widths sampled at 14 gates/decade.

Prior to the training of a DL method, inputs and the corresponding target outputs are normalized. Each resistivity model $\mathbf{m}$ is normalized, where the logarithmic variations in the model parameters can take both positive and negative values.

$$\mathbf{m}_n = \log_{10}(\mathbf{m}) - \frac{\mu[\log_{10}(m_{\max}) + \log_{10}(m_{min})]}{2}, \tag{2}$$

where $m_{min}$ and $m_{max}$ are the minimum and maximum resistivity values in the training data set of S-RMD, and $\mu$ is the mean.

The target outputs, i.e. $\frac{d\mathbf{B}}{dt}$, are normalized by:

$$\frac{d\mathbf{B}_n}{d\mathbf{t}} = \frac{\frac{d\mathbf{B}}{dt} - \mu[\frac{d\mathbf{B}}{dt}])}{\sigma[\frac{d\mathbf{B}}{dt}]}, \tag{3}$$

where $\mu$ is the mean and $\sigma$ is the standard deviation of each data point in the training data set.

We use a simple DL method where a fully-connected feed-forward neural network is utilized with two hidden layers having 384 neurons each. The hyperbolic tangent function is used as an activation function between the hidden layers and the full-batch scaled conjugate algorithm is used for backpropagation. The loss function for training is the sum of squared errors with

a regularization term consisting of the mean of sum of squares of the network weights and biases. The network configuration used here is based on our previous results (Asif et al., 2021b; Asif et al., 2022b). We also apply an early stopping criterion to





ensure that the training stops when validation loss starts to increase. The validation set for the early stopping criteria comprises of 70,000 models from S-RMD, which are excluded from the training set. Once the network is trained, it can be used for evaluation purposes. The evaluation metric for our baselines is the percentage relative error, $RL_P$, defined in Eq. (3), which
effectively deals with the large dynamic range and patterns of TEM data.

$$RL_P = \frac{(dB/dt)_{DL} - (dB/dt)_N}{(dB/dt)_N} \times 100\% \,, \tag{3}$$

where $(dB/dt)_{DL}$ is the output of the DL method and $(dB/dt)_N$ is the numerically computed forward response.

### 4.2 Surrogate forward modelling results

To test the performance of our DL method trained on S-RMD, we use 697 resistivity models inverted from field data from a
survey conducted in Søften, a region in Denmark. The minimum and maximum resistivity values in the test dataset are 3.9 Ωm and 127.1 Ωm respectively. The forward responses of the field inverted resistivity models are calculated numerically to compare it with the output of our DL method. Since the output of our DL algorithm are the normalized forward responses, it is de-normalized to raw data values by manipulating Eq. (3). For a relative comparison, we train another DL network with the same configuration using the initial von Kármán resistivity models. The comparison to the initial von Kármán resistivity
models also allow us to examine the effect of the *forward/inversion* process, as described in section 2, in the generation of the RMD.

Figure 3 shows the performance comparison of the trained DL networks based on the evaluation metric in Eq. (3) against the forward responses of 697 resistivity models from the Søften survey. Figure 3(a) shows the distribution of $RL_P$ of the DL network trained on S-RMD. We also show the accuracy performance of the DL network trained on von Kármán resistivity
models. It is evident that the network trained on S-RMD results in lower errors as compared to the network trained on von Kármán resistivity models. An improvement of 6% is achieved for the data points within half a percent relative error.

We also show the cumulative distribution of the $RL_P$ for the network trained on S-RMD and on von Kármán models in Figure 3(b). A maximum of 9% improvement in accuracy is achieved for the network is trained on the S-RMD as compared to the von Kármán models. The increased accuracy is achieved only by using an appropriate dataset for training. The prediction
accuracy can be improved with different data pre-processing, network configurations, loss functions, etc. while using the same training dataset to allow consistency in benchmarking of DL algorithms. It is also important that a balance between the prediction performance and computational efficiency is maintained. As such, the computational time for the forward pass of the proposed network configuration can serve as a baseline for time comparison.



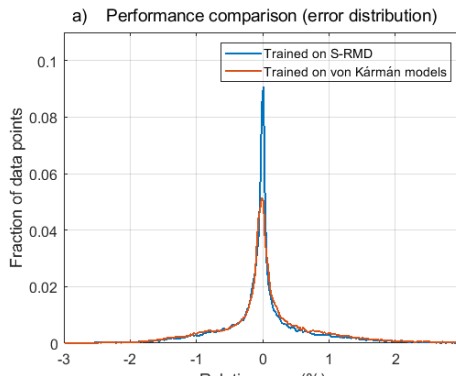 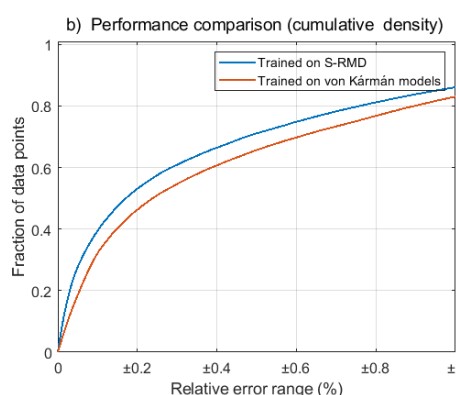


**Figure 3: Performance of the networks trained on S-RMD and von Kármán resistivity models. (a) RL$_P$ distribution. (b) Cumulative distribution of RL$_P$.**

Figure 4 show a visual comparison of a numerical forward response against the forward response from our DL networks for a resistivity model. It is evident from Figure 4 that the forward response from the network trained on S-RMD is more accurate

than the network trained on von Kármán models.

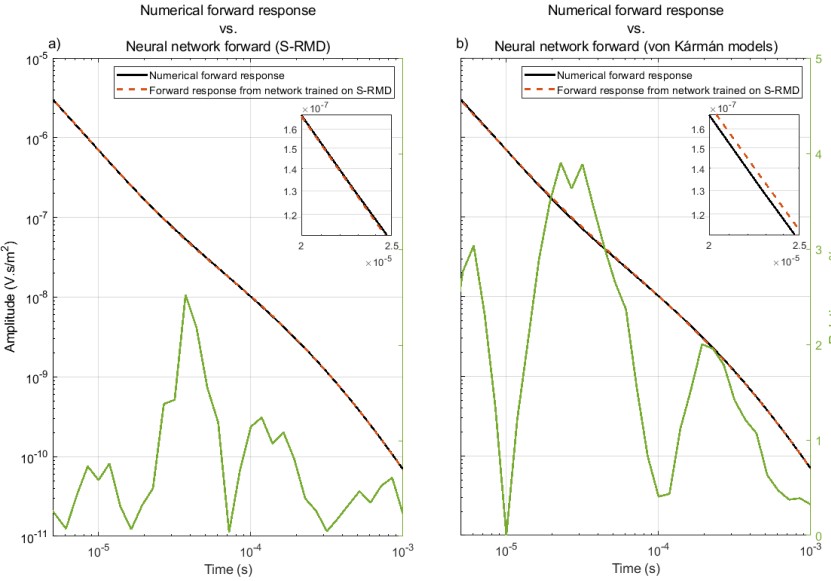

**Figure 4: Comparison of performance of the networks trained on S-RMD and von Kármán resistivity models with a numerical forward response from the test set. The forward responses are shown only within the time range of tTEM data., and the inset shows the forward response from 20 µs to 25 µs (a) Numerical forward response vs. the forward response from the network trained on S-RMD. (b) Numerical forward response vs. the forward response from the network trained on von Kármán models.**




## 5 Discussion

The DOI for a given TEM system strongly depends on the underlying resistivity model. Therefore, stating a single DOI value for a given TEM system is not appropriate. A single exploration depth, depth of investigation or a similar value stated by the instrument manufacturers will often be an optimistic one. For TEM systems with short transmitter current turn-off, the early

time data points provide the near surface resolution, while the late time data points strongly control the DOI for a given resistivity model. The transmitter moment and the background noise level also influence the DOI, but these factors are not considered in our case, since we have assumed a uniform data uncertainty in the forward and inversion process. The three RMD spans different TEM systems and resolutions. Therefore, for a particular TEM system, one should pick the RMD that has a similar resolution as the underlying generic TEM system. This is best evaluated by matching the time interval of the data

for the particular TEM system to the data time interval (data time start/end in Table 3) for the generic TEM system.

In Table 4, we list some examples of the compatibility of our RMD with some well-known TEM systems. Despite the I-RMD and D-RMD are compiled for generic airborne-system, the I-RMD and D-RMD are also appropriate for ground-based TEM systems since the simulated flight altitude of 40 m does not lead to a drastic change in the vertical resolution.

| System | Resistivity model database | | |
|---|---|---|---|
|  | S-RMD | I-RMD | D-RMD |
| EQUATOR (Karshakov et al., 2017) | ✓ | | |
| tTEM (Auken et al., 2018) | ✓ | | |
| MEGATEM (Smith et al., 2003) | | ✓ | |
| AEROTEM (Balch et al., 2003) | | ✓ | |
| SkyTEM (Sørensen and Auken, 2004) | | ✓ | ✓ |
| GEOTEM (Smith, 2010) | | ✓ | ✓ |
| SPECTREM$^{PLUS}$ (Leggatt et al., 2000) | | | ✓ |

**Table 4: Examples of RMD compatibility for some TEM systems.**

Since FEM and TEM systems follow the same laws of physics, the RMD is also applicable for many FEM systems, despite the generic EM system in the forward/inversion process mimics the TEM systems. In general, the FEM systems have a shallower DOI than that of the TEM systems, hence, the S-RMD is best suited for FEM systems. An alternative to the DL-RMD is to generate the resistivity model realizations by following the described methodology for the specific EM system by using the von Kármán models provided (ref to database). This will ensure a 100% match between resolution, DOI, etc. in the

model domain compared to sensitivity in the EM data domain.





Despite the initial von Kármán models with super-imposed layering, the resistivity models in the RMD have a pronounced vertical smooth behavior due to the minimum structure (smooth) regularization scheme (Viezzoli et al., 2008) used in the inversion phase. Applying another regularization scheme in the inversion phase, e.g., minimum support norm (Vignoli et al., 2015) or using a few layer model discretization with no vertical regularization, one could compile a resistivity model database

with different appearances. For our RMD, we chose the minimum structure regularization scheme, since it is commonly used for inverting airborne and ground-based EM data. It is important to point out that a TEM data curve itself does not hold information about whether subsurface boundaries are smooth or sharp. As such, both smooth and a sharp-layered model will explain the recorded data equally good in most cases. With our approach of compiling resistivity models, we have tried to avoid the inclusion of models with different smooth/sharp behavior that results in identical or close to identical forward data

responses (equivalent models).

The RMD is generated in the resistivity range of 1-2000 Ωm which covers most of the geological settings taking into account the EM mapping capability in high resistivity range. The resistivity limit of 2000 Ωm was chosen since EM methods have no or very little sensitivity in the high resistivity range, since high resistivity materials (e.g. granite, basalt, glacier ice, etc.) produces EM signal below the detection level. Despite the 2000 Ωm limit, the resistivity distribution of the models in the RMD

is slightly skewed towards lower resistivities due to the limited sensitivity of the EM method to high resistivity values. A slight bias towards lower resistivity values may affect the performance of a DL method for high resistive models. However, even if an actual subsurface model is represented by a high resistive model, it is expected that any TEM method would have difficulty in resolving such a model. The RMD also has a limitation in the low resistivity end, e.g., in settings with seawater and saltwater intrusion, which can result in subsurface materials with resistivity values below 1 Ωm.

Since the 1-D models of the RMD hold resistivity variations in one dimension (vertical) only, they cannot be used for calculating 2-D or 3-D EM-responses. Examples of geological settings where 1-D approach would be inappropriate include steep dipping geological structures, thin sheets mineralization, mapping close to or on the shoreline, or areas with strong topographical variations. However, one could apply the same methodology to compile a 2-D or 3-D resistivity database. In this case, one would generate the initial von Kármán models as 2-D section or 3-D volumes, and use a 2D or 3D forward and

inversion process, which of course would be much more computationally expensive compared to the 1-D case.

A network trained on random models may result in lower accuracy performance as compared to the network trained on von Kármán models. Due to the geological plausible nature of the von Kármán models, the network trained on such models still result in decent performance accuracy. However, a substantial improvement in accuracy is achieved when the geophysically resolvable models are employed for the network training. The accuracy performance of the DL methods can be further

improved by employing e.g. state-of-the-art convolutional neural networks. Such networks can learn complex patterns from



simple features. However, the RMD provided in this study opens up the possibility to explore more DL frameworks, and have reliability and consistency in performance comparisons.

## 6 Code and data availability

The DL-RMD is freely available at https://doi.org/10.5281/zenodo.7260886 (Asif et al., 2022a) and a ready to run demo code
in python jupyter notebook that uses the network trained on S-RMD and reproduce the results of this paper is available at https://github.com/rizwanasif/DL-RMD.

The EM modelling code "AarhusInv" (Auken et al., 2015) used in this study to generate EM forward responses is freely available to researchers for non-commercial activities. The details are available at https://hgg.au.dk/software/aarhusinv.

## 7 Conclusion

We have presented at methodology for compiling a geophysically constrained subsurface resistivity model database for applications related to electromagnetic data. We generated three 1-D resistivity databases, discretized to depths of 120 m, 350 m, and 500 m in the resistivity range of 1-2000 Ωm, hence covering various ground-based and airborne frequency-domain and time-domain electromagnetic systems and most of the geological settings. The upper resistivity limit of the model database is satisfactory as the electromagnetic methods have limitations for high resistivity, however, the model database has limitations
in the low resistivity for subsurface materials below 1 Ωm that may occur in some cases. Additionally, the database holds 1-D models and therefore inherit the limitations of 1-D electromagnetic modelling.

The example included using the RMD and DL for surrogating TEM forward modelling shows that high accuracy can be obtained with the RMD. Furthermore, the example shows that the forward/inversion steps in the generation of the RMD leads to a significantly increased performance in the forward modelling.

Despite some limitations, the generated resistivity model database is a well-organized database, which empowers the geoscience community to have consistency and credibility in the development of deep learning methods for many tasks including surrogating forward modelling, inverse modelling, data de-noising, automatic data processing, etc. Therefore, we urge the geophysical community to utilize the presented database to develop and investigate different network configurations, data pre-processing strategies, loss functions, etc. while using the presented model database to allow consistency in
benchmarking deep learning algorithms. The RMD has already proven valuable in significantly improving the accuracy of neural networks for the forward modelling of electromagnetic data.



**Financial support**

This work has been supported by the Innovation Fund Denmark (IFD) under the projects "MapField" (grant no. 8055-00025B), and "SuperTEM" (grant no. 0177-00085B).

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
