# Peer review of "DL-RMD: A geophysically constrained electromagnetic resistivity model database for deep learning applications"

_Earth System Science Data, 2022_

## Author Comment (AC1)

**General comments:**

This manuscript proposed a standardized dataset for deep learning-based electromagnetic methods. The database is geophysically constrained, which produces good accuracy performance and satisfactory generalization and consistency. Overall the paper is very well written, and the data shows its high readiness for the community. I would recommend it get published before some of my concerns are addressed.

**In response of general comments:**

The authors would like to thank you for your valuable time and positive evaluation of the paper under consideration. We agree with the suggestions, and they will improve the manuscript.

**Specific comments:**

**1a. Evaluation (section 4): This study mainly employed the proposed dataset to train a deep-learning (DL) model and surrogate the forward modeling process and demonstrated that this dataset shows its great performance. The assessment method is rational; however, this section needs additional result comparison with previous relevant DL-based studies. By comparing this proposed data with other DL studies that used limited input data, the authors may demonstrate this proposed database can be treated as the benchmark. Otherwise, it is only another DL experiment for improving the computation efficiency.**

 **In response of comment 1a:**

In order to make comparison with existing relevant studies, we generate random resistivity models similar to the data set generation approach used in several deep learning studies (Colombo et al., 2021b; Moghadas, 2020; Moghadas et al., 2020; Noh et al., 2020; Puzyrev and Swidinsky, 2021; Qin et al., 2019; Wu et al., 2021b) mentioned in Table 1. The data set generation approach for each of the above-mentioned deep learning studies differ from each other, however, the commonality in all the above-mentioned methods is that the resistivity values for each layer is chosen randomly. For example, Colombo et al. (2021b) generate 5-layer resistivity models with a random combination of resistivity and thickness; Moghadas (2020) generates 12-layer resistivity models with fixed layer thicknesses increasing logarithmically down to 6 meters; Moghadas et al. (2020) generate subsurface models having 12 layers with logarithmically increasing thickness down to 10 m depth; Noh et al. (2020) used 1 to 3 layer models; Puzyrev and Swidinsky (2021) generate resistivity models having 50 layers; and Wu et al. (2021b) generated resistivity models with number of layers randomly set between 1 and 20 with the bottom depth at 1000 m.

Therefore, to keep the network configuration same and to have same level of modelling complexity for a fair comparison; the number of layers, depth discretization, and the number of models will be kept the same as used to train other networks in our study. However, the resistivity values for each layer are picked randomly from a log-uniform distribution to compare with the deep learning studies that employ random resistivity models. The relevant text in the manuscript will be revised and Figure 3 and Figure 4 will also be updated after obtaining the results from random resistivity models.

**1b. Besides, please try to find some weaknesses in the training data utilized by previous DL studies and demonstrate your progress on it after comparison. For example, the introduction is well written but those training sets are not involved after then. Benchmark is a strong word that requires more comprehensive assessment and evidence.**

**In response of comment 1b:**

Some of the weaknesses in the training dataset utilized by other methods will be discussed in the revised manuscript. We also discuss the similarity of von Kármán models to the methods that uses pseudorandom resistivity models in some other deep learning studies and give the rationale behind improved performance using the proposed database.

**1c. The assessment section (section 4) needs to provide additional quantified comparisons and descriptions for the figures rather than just some summaries.**

**In response of comment 1c:**

The description of the figures and additional quantitative comparison will be included in the revised manuscript.

**2. References in Table 1 should be Qin et al. (2019) rather than (Qin et al. 2019), and generally, the table caption is above the table.**

**In response of comment 2:**

References in Table 1 will be formatted appropriately and captions for the tables will be moved above for all the tables in the manuscript.

**3. The dataset is formatted as txt, which caused the code reading speed very slow.**

**In response of comment 3:**

We have chosen to use the txt format as it is easily readable. As the data from the txt file needs to be read only once for a particular algorithm, the authors are of the opinion to keep it as it is.

---

## Author Comment (AC2)

**General comments:**

This paper submitted by Asif et al. describes a geophysically constrained subsurface resistivity model database for electromagnetic systems in a deep learning context. Such datasets and associated analyses are valuable in applying deep learning methods to geophysical applications. So the paper and associated database should be of interest to engineers and those interested in applying deep learning to electromagnetic methods. The manuscript is overall well organized and written, but there are still some shortcomings that need to be addressed before it can be accepted for publication.

**In response of general comment:**

      The authors would like to thank you for your valuable time and positive evaluation of the paper under consideration. We will revise the manuscript to address the issues raised.

**Specific comments:**

**1. Please enhance the description of data processing in the revised manuscript. This will be important for a full understanding of this dataset.**

**In response of comment 1:**

      It is a bit unclear which part of the manuscript the reviewer is pointing at. Therefore, we will expand the manuscript where ever some data is processed, which is either processing of the EM forward data for inversion to obtain EM resolvable models, or the processing of field data which is used for comparison.

**2. My main suggestion relates to Section 4 of this paper. As you stated, the deep learning resistivity model database (DL-RMD) presented in this paper can provide uniformity in benchmarking for DL methods in EM. But Section 4 doesn't really provide a clear description of the great performance of this dataset. It would be better to compare it with other DL studies that have been published and listed in the Introduction.**

**In response of comment 2:**

      In order to make comparison with existing relevant studies, we generate random resistivity models similar to the data set generation approach used in several deep learning studies (Colombo et al., 2021b; Moghadas, 2020; Moghadas et al., 2020; Noh et al., 2020; Puzyrev and Swidinsky, 2021; Qin et al., 2019; Wu et al., 2021b) mentioned in Table 1. The data set generation approach for each of the above-mentioned deep learning studies differ from each other, however, the commonality in all the above-mentioned methods is that the resistivity values for each layer is chosen randomly. For example, Colombo et al. (2021b)

generate 5-layer resistivity models with a random combination of resistivity and thickness; Moghadas (2020) generates 12-layer resistivity models with fixed layer thicknesses increasing logarithmically down to 6 meters; Moghadas et al. (2020) generate subsurface models having 12 layers with logarithmically increasing thickness down to 10 m depth; Noh et al. (2020) used 1 to 3 layer models; Puzyrev and Swidinsky (2021) generate resistivity models having 50 layers; and Wu et al. (2021b) generated resistivity models with number of layers randomly set between 1 and 20 with the bottom depth at 1000 m.

Therefore, to keep the network configuration same and to have same level of modelling complexity for a fair comparison; the number of layers, depth discretization, and the number of models will be kept the same as used to train other networks in our study. However, the resistivity values for each layer are picked randomly from a log-uniform distribution to compare with the deep learning studies that employ random resistivity models. The relevant text in the manuscript will be revised and Figure 3 and Figure 4 will also be updated after obtaining the results from random resistivity models.

**3. Table 1 – It needs to be greatly improved. The table caption is generally above the table. The references in the Table should be changed to "Wu et al. (2021a) ". The table caption should be concise but descriptive.**

**In response of comment 3:**

References in Table 1 will be formatted as mentioned and the captions for the tables will be moved above for all the tables in the revised manuscript. The caption was Table 1 will also be revised to be concise.

**4. Equations – Please check the writing form (e.g. $C_0$). Equation 2 – Suggest revising "log10" to "lg".**

**In response of comment 4:**

The equations will be checked thoroughly and corrected wherever necessary, e.g. Equation 2. However, due to some issues with MS Word equation tools, the writing form $C_0$ can not be corrected. However, the authors will keep it in mind during the proof-read to ensure that it is corrected.

Additionally, "lg" is not a commonly used term and could create confusion among the readers. Therefore, the authors are of the opinion to keep "$\log_{10}$" for clarity.

**5. Figure 4 – Poor quality. It would be better to provide additional descriptions for the figures rather than just summaries. For this figure, only one sentence was used to describe.**

**In response of comment 5:**

Figure 4 will be updated and additional descriptions will be included for Figure 3 and Figure 4.

**6. There are lots of abbreviations used in this manuscript, it would be better to add an Appendix. Abbreviations in the title should be avoided. The phrase "depth of investigation" is abbreviated as the "DOI", this abbreviation is not recommended.**

**In response of comment 6:**

We will revise the manuscript and replace the abbreviations of depth of investigation (DOI) and deep learning (DL) to their full forms throughout the manuscript for clarity. Then, there will remain only seven abbreviations used in the manuscript, i.e. EM, TEM, FEM, DL-RMD, S-RMD, I-RMD and D-RMD. Therefore, the authors are of the opinion not to add an Appendix.

---

## Author Response (AR1)

**Earth System Science Data**

Ref: essd-2022-345

Title: DL-RMD: A geophysically constrained electromagnetic resistivity model database for deep learning applications

Dear editor, topical editor, and reviewers,

The authors would like to thank you for your valuable time and constructive comments for the paper under consideration. As per the suggestions of the reviewers, we have carefully revised the manuscript and made changes where necessary.

The point-by-point reply to the comments is found on the following pages. The reproduced text from the manuscript is highlighted in gray and the changes made are marked in red.

On behalf of the authors

**Anonymous Referee #1**

**General comments:**

This manuscript proposed a standardized dataset for deep learning-based electromagnetic methods. The database is geophysically constrained, which produces good accuracy performance and satisfactory generalization and consistency. Overall the paper is very well written, and the data shows its high readiness for the community. I would recommend it get published before some of my concerns are addressed.

**In response of general comments:**

The authors would like to thank you for your valuable time and positive evaluation of the paper under consideration. We have carefully revised the manuscript to address the raised concerns. The point-by-point reply to the comments and the reproduced text from the revised manuscript is found on the following pages. The reproduced text from the revised manuscript is highlighted in gray and the changes made in the manuscript are marked in red.

**Specific comments:**

**1a. Evaluation (section 4): This study mainly employed the proposed dataset to train a deep-learning (DL) model and surrogate the forward modeling process and demonstrated that this dataset shows its great performance. The assessment method is rational; however, this section needs additional result comparison with previous relevant DL-based studies. By comparing this proposed data with other DL studies that used limited input data, the authors may demonstrate this proposed database can be treated as the benchmark. Otherwise, it is only another DL experiment for improving the computation efficiency.**

**In response of comment 1a:**

We generate random resistivity models similar to the data set generation approach used in several deep learning studies (Colombo et al., 2021b; Moghadas, 2020; Moghadas et al., 2020; Noh et al., 2020; Puzyrev and Swidinsky, 2021; Qin et al., 2019; Wu et al., 2021b) mentioned in Table 1. The data set generation approach for each of the above-mentioned deep learning studies differ from each other, however, the commonality in all the above-mentioned methods is that the resistivity values for each layer is chosen randomly. For example, Colombo et al. (2021b) generate 5-layer resistivity models with a random combination of resistivity and thickness; Moghadas (2020) generates 12-layer resistivity models with fixed layer thicknesses increasing logarithmically down to 6 meters; Moghadas et al. (2020) generate subsurface models having 12 layers with logarithmically increasing thickness down to 10 m depth; Noh et al. (2020) used 1 to 3 layer models; Puzyrev and Swidinsky (2021) generate resistivity models having 50 layers; and

Wu et al. (2021b) generated resistivity models with number of layers randomly set between 1 and 20 with the bottom depth at 1000 m.

Therefore, to keep the network configuration same and to have same level of modelling complexity for a fair comparison; the number of layers, depth discretization, and the number of models have been kept the same as used to train other networks in our study. However, the resistivity values for each layer are picked randomly from a log-uniform distribution to compare with the deep learning studies that employ random resistivity models. The relevant text has been revised and Figure 3 and Figure 4 have been updated after obtaining the results from random resistivity models. In essence, whole of *Section 4.2: Surrogate forward modelling results* has been modified, and is reproduced below:

[revised manuscript text omitted]

Figure 4 show a visual comparison of a numerical forward response against the forward response from the trained networks for one of the resistivity models from the Søften survey. It is evident from Figure 4 that the forward response from the network trained on S-RMD is the most accurate and has a maximum relative error of 1.4% for the data point at ~72 µs (see Figure 4a). The highest error for the forward response from the network trained on von Kármán models is observed to be 2.5% for the data point at ~160 µs as shown in Figure 4(b). The forward response from the network trained on random models results in the worst accuracy performance and results in a maximum error of 22.3% for the data point at 100 µs (see Figure 4c).

[Figure]

**Figure 2: Comparison of performance of the networks trained on S-RMD, von Kármán and random resistivity models with a numerical forward response from the test set. The forward responses are shown only within the time range of tTEM data, and the inset shows the forward response from 16 μs to 20 μs (a) Numerical forward response vs. the forward response from the network trained on S-RMD. (b) Numerical forward response vs. the forward response from the network trained on von Kármán models. (c) Numerical forward response vs. the forward response from the network trained on random resistivity models.**

**1b. Besides, please try to find some weaknesses in the training data utilized by previous DL studies and demonstrate your progress on it after comparison. For example, the introduction is well written but those training sets are not involved after then. Benchmark is a strong word that requires more comprehensive assessment and evidence.**

**In response of comment 1b:**

Some of the weaknesses in the training dataset utilized by other methods is now discussed in the first paragraph of *Section 5: Discussion*. In the same paragraph, we also discuss the similarity of von Kármán models to the methods that uses pseudorandom resistivity models in some other deep learning studies and give the rationale behind improved performance using the proposed database. The revised paragraph is reproduced below:

The network trained on random resistivity models results in poor accuracy performance as many of the resistivity models in the training dataset are geologically unrealistic. The complex unrealistic resistivity structures in the randomly generated training models would result in forward responses similar to the ones obtained from simpler resistivity models which further decreases the quality of the training dataset. The von Kármán models may be

considered as pseudorandom resistivity models where the resistivity structure of the models have geological realistic nature as it considers multiple correlation lengths with stochastic nature resembling geological processes. Due to the geological nature of the von Kármán models, the network trained on such models result in decent performance accuracy. However, the network trained on von Kármán models has lower accuracy performance as compared to the network trained on S-RMD where the resolution capability of the EM method has been taken into account which results in resistivity structures resolvable by the EM method.

**1c. The assessment section (section 4) needs to provide additional quantified comparisons and descriptions for the figures rather than just some summaries.**

**In response of comment 1c:**

The description of the figures and additional quantitative comparison has now been included in *Section 4.2: Surrogate forward modelling results*. Please see the author's response to comment 1a in the above pages.

**2. References in Table 1 should be Qin et al. (2019) rather than (Qin et al. 2019), and generally, the table caption is above the table.**

**In response of comment 2:**

References in Table 1 have been formatted as mentioned and the captions for the tables have been moved above for all the tables in the manuscript.

**3. The dataset is formatted as txt, which caused the code reading speed very slow.**

**In response of comment 3:**

We have chosen to use the txt format as it is easily readable. As the data from the txt file needs to be read only once for a particular algorithm, the authors are of the opinion to keep it as it is.

**Anonymous Referee #2**

**General comments:**

This paper submitted by Asif et al. describes a geophysically constrained subsurface resistivity model database for electromagnetic systems in a deep learning context. Such datasets and associated analyses are valuable in applying deep learning methods to geophysical applications. So the paper and associated database should be of interest to engineers and those interested in applying deep learning to electromagnetic methods. The manuscript is overall well organized and written, but there are still some shortcomings that need to be addressed before it can be accepted for publication.

**In response of general comment:**

The authors would like to thank you for your valuable time and positive evaluation of the paper under consideration. We have revised the manuscript to address the issues raised. The point-by-point reply to the comments is found on the following pages. The reproduced text from the manuscript is highlighted in gray and the changes made are marked in red.

**Specific comments:**

**1. Please enhance the description of data processing in the revised manuscript. This will be important for a full understanding of this dataset.**

**In response of comment 1:**

It is a bit unclear which part of the manuscript the reviewer is pointing at. Therefore, we have expanded the manuscript where ever some data is processed, which is either processing of the EM forward data for inversion to obtain EM resolvable models, or the processing of field data which is used for comparison.

The paragraph discussing the processing of the EM forward data for inversion to obtain EM resolvable models in *Section 2: Methodology* has been revised and is reproduced below:

> Prior to the EM forward calculation, the von Kármán models are re-discretized to 90-layers for faster forward computation and easier handling. The top layer thickness and depth to the last layer boundary for the re-discretized layers have been detailed in Table 3 for three generic EM systems having different depth of investigations (see Section 3 for further details). For the forward calculation, the geometric mean of the last 5 meters of the re-discretized von Kármán models is assigned to the last model layer that continues to infinity depth. In order to avoid making assumptions on the acquisition conditions, specific instrument setup, etc., the calculated forward data are pragmatically assigned a uniform uncertainty of 5% to take noise into account and is inverted with a 30-layer model with a minimum structure (smooth) regularization scheme (Viezzoli et al., 2008). The layer thicknesses for the 30-layer models are fixed and have also been listed in Table 3. The red model curves in Figure 1 are the resistivity models after the forward and inversion process, and are the models that enter the DL-RMD. As seen from Figure 1, the von Kármán models hold structures that are not resolved by the inverted resistivity models, so the models obtained after the forward and inversion process results in structures resolvable by the EM method. A total of ~95% of the inverted

resistivity models explain (fits) the forward data within the assumed data uncertainty. In other words, the inverted models are explaining the more complex von Kármán models to a very high degree.

Additionally, the text discussing the processing and inversion of field data used for comparison in *Section 4.2: Surrogate forward modelling results* has been modified, and is reproduced below:

> To test the performance of our DL method trained on S-RMD, we use 697 resistivity models inverted from field data from a survey conducted in Søften, a region in Denmark. The data processing and inversion of the field data follows Auken et al. (2018), which covers averaging, anomaly detection, manual inspection, etc. on the data. The minimum and maximum resistivity values in the test dataset are 3.9 Ωm and 127.1 Ωm respectively. The forward responses of the field inverted resistivity models are calculated numerically to compare it with the output of our DL method.

**2. My main suggestion relates to Section 4 of this paper. As you stated, the deep learning resistivity model database (DL-RMD) presented in this paper can provide uniformity in benchmarking for DL methods in EM. But Section 4 doesn't really provide a clear description of the great performance of this dataset. It would be better to compare it with other DL studies that have been published and listed in the Introduction.**

**In response of comment 2:**

We generate random resistivity models similar to the data set generation approach used in several deep learning studies (Colombo et al., 2021b; Moghadas, 2020; Moghadas et al., 2020; Noh et al., 2020; Puzyrev and Swidinsky, 2021; Qin et al., 2019; Wu et al., 2021b) mentioned in Table 1. The data set generation approach for each of the above-mentioned deep learning studies differ from each other, however, the commonality in all the above-mentioned methods is that the resistivity values for each layer is chosen randomly. For example, Colombo et al. (2021b) generate 5-layer resistivity models with a random combination of resistivity and thickness; Moghadas (2020) generates 12-layer resistivity models with fixed layer thicknesses increasing logarithmically down to 6 meters; Moghadas et al. (2020) generate subsurface models having 12 layers with logarithmically increasing thickness down to 10 m depth; Noh et al. (2020) used 1 to 3 layer models; Puzyrev and Swidinsky (2021) generate resistivity models having 50 layers; and Wu et al. (2021b) generated resistivity models with number of layers randomly set between 1 and 20 with the bottom depth at 1000 m.

Therefore, to keep the network configuration same and to have same level of modelling complexity for a fair comparison; the number of layers, depth discretization, and the number of models have been kept the same as used to train other networks in our study. However, the resistivity values for each layer are picked randomly from a log-uniform distribution to compare with the deep learning studies that employ random resistivity models. The relevant text has been revised and Figure 3 and Figure 4 have been updated

after obtaining the results from random resistivity models. In essence, whole of *Section 4.2: Surrogate forward modelling results* has been modified, and is reproduced below:

[revised manuscript text omitted]

Figure 4 show a visual comparison of a numerical forward response against the forward response from the trained networks for one of the resistivity models from the Søften survey. It is evident from Figure 4 that the forward response from the network trained on S-RMD is the most accurate and has a maximum relative error of 1.4% for the data point at ~72 μs (see Figure 4a). The highest error for the forward response from the network trained on von Kármán models is observed to be 2.5% for the data point at ~160 μs as shown in Figure 4(b). The forward response from the network trained on random models results in the worst accuracy performance and results in a maximum error of 22.3% for the data point at 100 μs (see Figure 4c).

[Figure]

**Figure 4: Comparison of performance of the networks trained on S-RMD, von Kármán and random resistivity models with a numerical forward response from the test set. The forward responses are shown only within the time range of tTEM data, and the inset shows the forward response from 16 μs to 20 μs (a) Numerical forward response vs. the forward response from the network trained on S-RMD. (b) Numerical forward response vs. the forward response from the network trained on von Kármán models. (c) Numerical forward response vs. the forward response from the network trained on random resistivity models.**

**3. Table 1 – It needs to be greatly improved. The table caption is generally above the table. The references in the Table should be changed to " Wu et al. (2021a) ". The table caption should be concise but descriptive.**

**In response of comment 3:**

References in Table 1 have been formatted as mentioned and the captions for the tables have been moved above for all the tables in the manuscript. The caption was Table 1 has also been revised to be concise.

**4. Equations – Please check the writing form (e.g. C₀). Equation 2 – Suggest revising "log10" to "lg".**

**In response of comment 4:**

The equations have been checked and corrected wherever necessary, e.g. Equation 2. However, due to some issues with MS Word equation tools, the writing form $C_0$ can not be corrected in this version. However, the authors will keep it in mind during the proof-read to ensure that it is corrected.

Additionally, "lg" is not a commonly used term and could create confusion among the readers. Therefore, the authors are of the opinion to keep "$\log_{10}$" for clarity.

**5. Figure 4 – Poor quality. It would be better to provide additional descriptions for the figures rather than just summaries. For this figure, only one sentence was used to describe.**

**In response of comment 5:**

Figure 4 has been updated and additional descriptions have been included for Figure 3 and Figure 4 in *Section 4.2: Surrogate forward modelling results*. Please see the authors' response to comment 4 for the updated text.

**6. There are lots of abbreviations used in this manuscript, it would be better to add an Appendix. Abbreviations in the title should be avoided. The phrase "depth of investigation" is abbreviated as the "DOI", this abbreviation is not recommended.**

**In response of comment 6:**

We have revised the manuscript and replaced the abbreviations of depth of investigation (DOI) to its full forms throughout the manuscript for clarity. Based on editor's suggestion, we retain the abbreviation DL for deep learning. Now, there remains only eight abbreviations used in the manuscript, i.e. EM, TEM, FEM, DL-RMD, S-RMD, I-RMD, D-RMD and DL. Therefore, the authors are of the opinion not to add an Appendix.

---

## Author Response (AR2)

**Earth System Science Data**

Ref: essd-2022-345

Title: DL-RMD: A geophysically constrained electromagnetic resistivity model database for deep learning applications

Dear editor, topical editor, and the editorial support assistance,

The authors would like to thank you for your valuable time and contributions during the review process of the paper accepted for publication titled above. As per the comment of the topical editor, we have carefully revised the manuscript for any grammatical errors.

Additionally, it has been mentioned file validation phase that the reference list includes works "in review". It is to inform that it is a misunderstanding which has been communicated earlier that work that is being pointed out is titled, "70 years of machine learning in geoscience in review" which has been published in 2020. "in review" is in the title of the paper. The authors were notified that a note will be added into the system to avoid the misunderstanding. However, the DOI of the said reference is included in this submission.

On behalf of the authors